# Excitation and inhibition in anterior cingulate predict use of past experiences

**Jacqueline Scholl[1]\*[†], Nils Kolling[1][†], Natalie Nelissen[2], Charlotte J Stagg[2], Catherine J Harmer[3][‡], Matthew FS Rushworth[1,4][‡]**

[1]Department of Experimental Psychology, University of Oxford, Oxford, United Kingdom; [2]Oxford Centre for Human Brain Activity, Department of Psychiatry, Oxford University, Oxford, United Kingdom; [3]Department of Psychiatry, University of Oxford, Oxford, United Kingdom; [4]Oxford Centre for Functional MRI of the Brain, University of Oxford, Oxford, United Kingdom

**\*For correspondence:** jacqueline. scholl@cantab.net

[†]These authors contributed equally to this work
[‡]These authors also contributed equally to this work

**Competing interests:** The authors declare that no competing interests exist.

**Abstract** Dorsal anterior cingulate cortex (dACC) mediates updating and maintenance of cognitive models of the world used to drive adaptive reward-guided behavior. We investigated the neurochemical underpinnings of this process. We used magnetic resonance spectroscopy in humans, to measure levels of glutamate and GABA in dACC. We examined their relationship to neural signals in dACC, measured with fMRI, and cognitive task performance. Both inhibitory and excitatory neurotransmitters in dACC were predictive of the strength of neural signals in dACC and behavioral adaptation. Glutamate levels were correlated, first, with stronger neural activity representing information to be learnt about the tasks' costs and benefits and, second, greater use of this information in the guidance of behavior. GABA levels were negatively correlated with the same neural signals and the same indices of behavioral influence. Our results suggest that glutamate and GABA in dACC affect the encoding and use of past experiences to guide behavior.

## Introduction

Dorsal anterior cingulate cortex (dACC) has a central role in reward-guided decision-making, behavioral adaptation, learning, and formation of task models (*Heilbronner and Hayden, 2016*; *Kolling et al., 2016a*; *Holroyd and Yeung, 2012*; *Khamassi et al., 2011*; *Ullsperger et al., 2014*). Recently dACC's role in health and disease has been underscored by findings that structural variability predicts a broad spectrum of mental illnesses (*Goodkind et al., 2015*). Most of our knowledge of dACC is based on measurements tied to neuronal firing such as human functional magnetic resonance imaging (fMRI) and animal recording studies or to investigations of loss of function after lesions and inactivation (*Kennerley et al., 2006*; *Amiez et al., 2006*). However, the neurochemical modulation and orchestration of dACC's role is largely unknown.

The importance of variation in neurotransmitter levels has recently become apparent in other frontal brain areas. For example ventromedial prefrontal cortex (vmPFC) has been linked to value-guided decisions (*Boorman et al., 2009*; *Rushworth et al., 2011*). Biophysical neural network models of decision-making in vmPFC (*Hunt et al., 2012*) predict that the inhibitory neurotransmitter gamma-aminobutyric acid (GABA) mediates the dynamics of the value comparison process. The predictions were born out in a study looking at the neurochemistry of this structure with magnetic resonance spectroscopy (MRS) (*Jocham et al., 2012*). Relatedly, levels of GABA in motor cortex (*Stagg et al., 2011*) and in the frontal eye field (*Sumner et al., 2010*) have been found to predict the speed of selection of responses and inhibition of incorrect responses to distractors respectively. In all three cases, neurotransmitter levels were predictive of the dynamics of the decision or selection process within different domains.

Here we use a similar approach to examine the relation between GABA and glutamate in dACC, fMRI-based indices of neural activity, and behavior. We relate these neurotransmitters to a key function of dACC that is quite distinct to the selection processes previously examined in MRS studies, namely the use of a task model to guide behavior based on past experience. More specifically, we hypothesized that if excitatory and inhibitory neurotransmitters in dACC determine the processing and use of information to form a model of the world (*O'Reilly et al., 2013*), or at least the task at hand, then measures of these neurotransmitters should relate to both behavioral and neural markers of this process (*Figure 1—figure supplement 1*).

## Results

We used MRS to obtain measures of the total amount of GABA and glutamate in 27 humans at rest in dACC (*Figure 1A and B*). Participants then performed a previously established multi-dimensional learning task (*Scholl et al., 2015*) during fMRI acquisition. Participants had to repeatedly choose between the same two options, based on the reward probabilities and the reward and effort magnitudes (i.e. requirement of a sustained effort) associated with each option. The reward probabilities changed randomly from trial to trial and were displayed to participants on each trial on the screen. By contrast the reward and effort magnitudes associated with each option had to be learnt from experience across trials (*Figure 1C and D*). The participants' goal was thus to choose options that would lead to the highest reward magnitude with the highest probability of being rewarded, but at the same time requiring the least effort. Participants performed the task well (*Figure 2*) after careful training.

Participants' performance can be described using a computational reinforcement-learning model (see *Figure 2—figure supplement 1* and *2*). This allows parsing a single behavior (choices on each trial) into different underlying components. Our hypothesis was that neurotransmitter levels in dACC should relate to how much participants used the learnt information or, in other words, a model of what choices are associated with high/low reward/effort magnitudes, to guide their choices (rather than just relying on the displayed probability information). This use of learnt information was captured by a single parameter in the model ($\gamma$, *Figure 2—figure supplement 1C*), which was independent from participants' other behavioral parameters (*Figure 2—figure supplement 2B*).

If the use of learnt information depends on the excitation/inhibition balance, we should find correlations between $\gamma$ and the neurotransmitters. Indeed, partial correlation analyses revealed that higher glutamate relative to GABA levels related to increased use of the learnt information ($\rho$=0.53, p=0.011). This effect was specific to the use of learnt information (*Figure 3—figure supplement 1*). When considering the effects of the two neurotransmitters separately, we found that both higher levels of glutamate ($\rho$=0.45, p=0.039) and lower levels of GABA ($\rho$=−0.43, p=0.05) were independently related to increased use of the learnt information (*Figure 3A*).

One way in which resting state glutamate/GABA levels could be linked to behavioral performance is through an impact on brain activity. To test this, we first identified brain areas that represented the information to be learnt (GLM1) at the time of learning. We identified activity in dACC and adjacent cortex (*Figure 3Bi*, x = 6, y = 32, z = 36, z-score = 3.62, cluster p-value=$5*10^{-5}$) and in other areas (*Table 1A*) as coding the information to be learnt as an inverse outcome value signal (relative reward outcome minus relative effort outcome) or, in other words, a signal related to the relative value of the alternative not chosen on the current trial. Such a signal has previously been noted in dACC and has been related to behavioral adaptation: decisions to maintain or change behavior in diverse contexts (*Shima and Tanji, 1998*; *Kolling et al., 2012*; *Stoll et al., 2016*; *Meder et al., 2016*; *Kolling et al., 2016b*). Other areas with different types of outcome-related activity are listed in *Table 1B*. Next, we examined whether variation in this neural signal was related to our behavioral measure of use of learnt information ($\gamma$, GLM2). Again, we found this to be the case in a partly overlapping dACC area (*Figure 3Bii* and *Table 1C*, x=−14, y = 24, z = 58, z-score = 3.44, cluster p-value=$2*10^{-5}$): participants with stronger neural representation of the information to be learnt in dACC were better at using the learnt information to guide their choices.

Finally, we tested whether neurotransmitter levels in dACC were related to the neural representation of the information to be learnt (GLM3). Indeed, we found that the strength of the representation of the information to be learnt in dACC correlated with the relative glutamate to GABA levels (*Figure 3Biii*, x = 4, y = 22, z = 40, z-score = 3.11, cluster p-value=0.039). This result was specific to

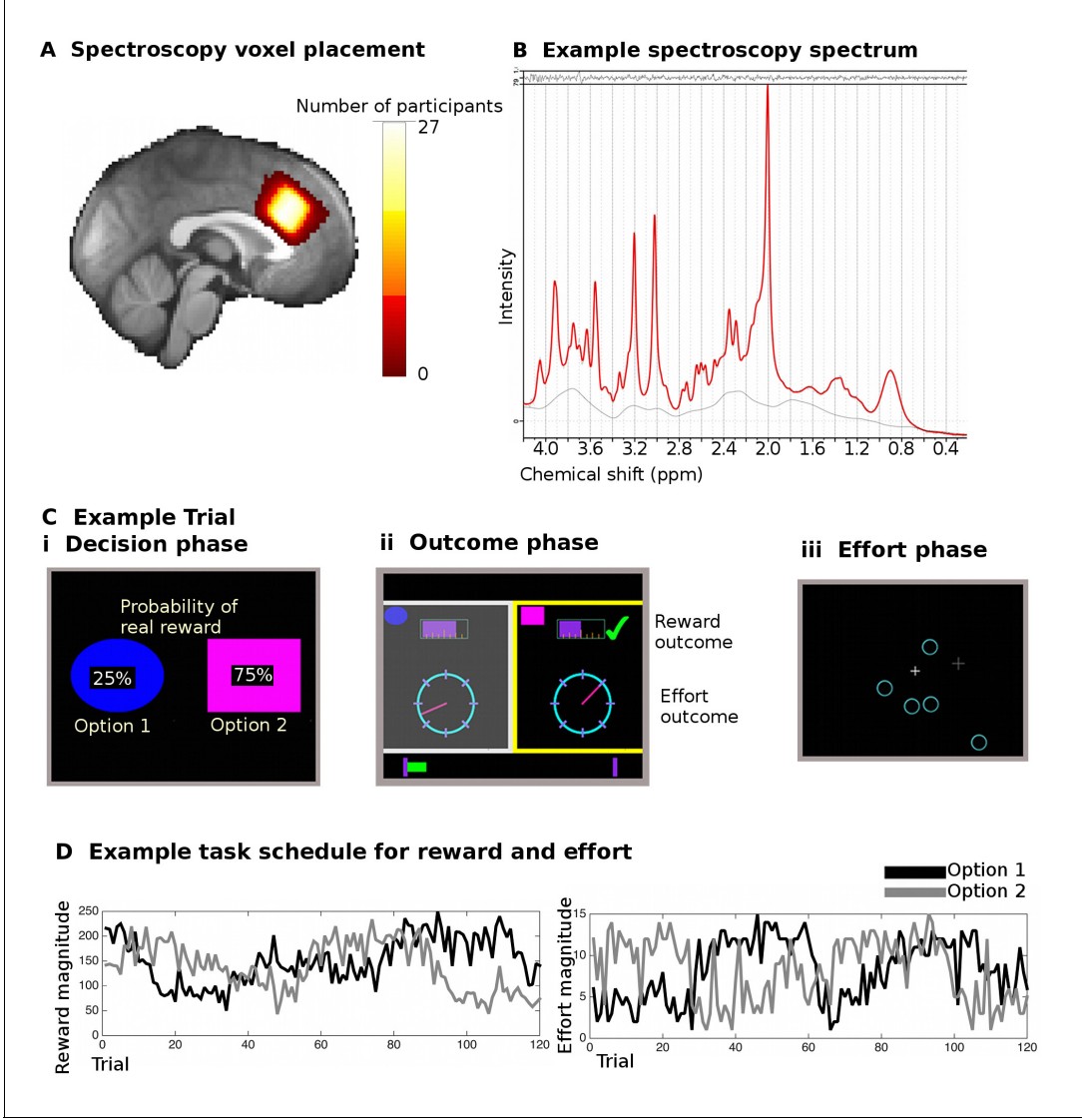

**Figure 1.** Spectroscopy measurements and task. (A) Spectroscopy voxels were placed in dACC. Cingulate sulcal morphology was used to guide voxel placement and this resulted in consistent positioning of the voxel in the same location in MNI space (white color indicates overlap in voxel position in all 27 participants). (B) Example spectroscopy spectrum from one participant. The fitted LCModel (red) is plotted overlaid on the actual data (black). The difference between the data and the model (residuals) is shown at the top and the baseline at the bottom. (C) Participants performed 240 trials of a reward- and effort-guided learning task. On each trial, participants were shown two options overlaid with the probability of receiving a reward for each choice (Ci). Participants chose between the options on the basis of the reward probabilities displayed on the screen and on the basis of reward and effort magnitudes learnt from experience on previous trials. After participants chose one option, they were shown feedback information for both the option they had chosen and the unchosen option (Cii). The reward magnitudes were shown as purple bars (top of the screen), the effort magnitude was indicated through the position of a dial on a circle. Whether the participant actually received a reward or not (because of the reward probability) was indicated through a tick mark (green) or a cross (red, not shown here). If participants received a reward, the chosen reward magnitude was added to a status bar at the bottom of the screen, which tracked participants' earning over the course of the experiment. Finally, participants exerted the effort associated with the chosen option in a final phase of the trial (Ciii). They had to exert an extended effort by responding to select targets that appeared on the screen over a period of time before the trial ended; the higher the trial's effort level, the more targets participants had to eliminate. (D) Example of reward magnitude and effort magnitude variation associated with the two options across the course of the 120 trials for one of the two sessions in the experiment.

The following figure supplement is available for figure 1:

**Figure supplement 1.** Illustration of the model of integration and use of learnt information and relationship between spectroscopy and brain activity.

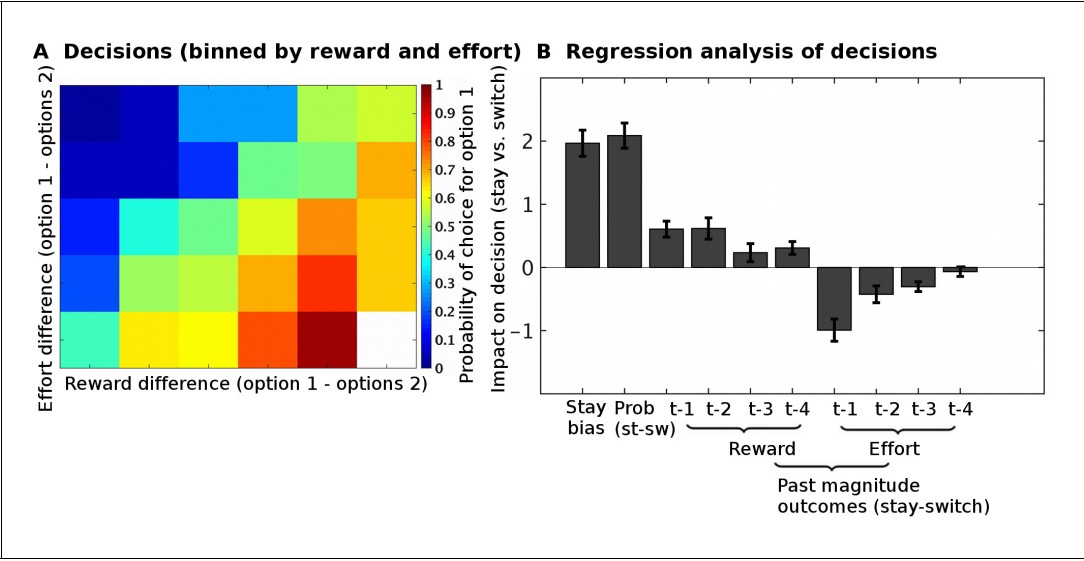

**Figure 2.** Task validation. (**A**) The choices of participants, between the two options, were guided by the reward and effort differences between the options (estimated from a Bayesian learning model): participants were more likely to choose options with higher predicted reward magnitudes and lower predicted effort magnitudes. (**B**) In a logistic regression analysis, we measured the impact of different factors on choices to either select the same option again as on the previous trial ('stay') or to instead select the alternative option ('switch'). The factors we included were: the reward probabilities ('prob') displayed to participants, the reward and effort magnitude outcomes on the past four trials (t-1, t-2, t-3, t-4). This regression showed that participants could use all the relevant information when making their decisions; they used the reward probabilities that were shown on the screen ($t_{(26)} = 10.5$, $p = 8*10^{-11}$), the past reward magnitudes (ANOVA across all four past trials, main effect: $F_{(1,25)} = 53.5$, $p = 1*10^{-7}$), and the past effort magnitudes (ANOVA across all four past trials, main effect: $F_{(1,25)} = 86.9$, $p = 1*10^{-9}$). Data in Figure2_SourceData.xlsx

The following source data and figure supplements are available for figure 2:

**Source data 1.** This table contains the regression coefficients for individual participants for the analysis shown in *Figure 2B*.

**Source data 2.** This table relates to *Figure 2—figure supplement 1*.

**Source data 3.** This table relates to *Figure 2—figure supplement 2*.

**Figure supplement 1.** Model simulation and validation.

**Figure supplement 2.** Model fit and parameter stability.

---

dACC; analogous analyses in other ROIs identified in the contrasts for learnt information (*Table 1A and B*) revealed no significant effects. These findings suggest that neurotransmitters in dACC are predictive of a behavior dependent on dACC and of fMRI-based measures of neural activity in dACC related to the same behavior.

## Discussion

We looked at the effects of neurotransmitter variation on dACC function. We found that differences in glutamate and GABA both related, firstly, to the strength of neural signals in dACC encoding the outcomes of decisions, i.e. the feedback information that should guide behavioral adaptation on future decisions. Secondly, the neurotransmitters also related to behavior, i.e. how well participants used this feedback information to guide future choices. Strikingly, we found opposing patterns of relationships for excitatory and inhibitory neurotransmitters: higher levels of glutamate and lower levels of GABA were linked to increased use of the learnt information.

Our findings are consistent with an emerging view of dACC in forming, updating and maintaining a model of the world and of behavioral strategies (*O'Reilly et al., 2013*; *Karlsson et al., 2012*; *Kolling et al., 2014*; *Wittmann et al., 2016*). In our paradigm, it was always advantageous to use

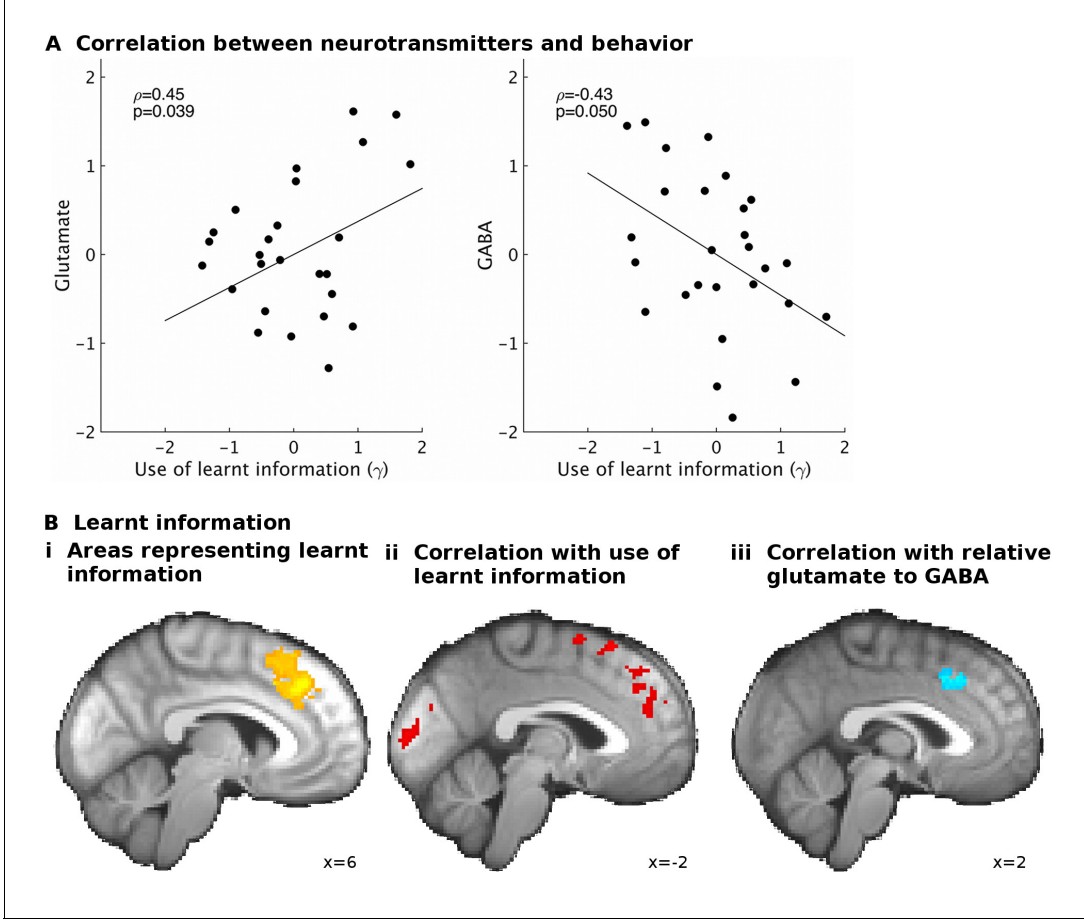

**Figure 3.** GABA and glutamate predict behavior and neural activity. (**A**) Participants with higher concentrations of glutamate (ρ=0.45, p=0.039) and lower concentrations of GABA (ρ=−0.43, p=0.05) in dACC were better able to use the learnt information (parameter γ from computational model) to guide their choices (the graphs illustrate partial correlations, i.e. the plotted values have been adjusted for covariates, see Materials and methods). (**Bi**) Neural activity in dACC was sensitive to the information to be learnt at the time of the outcome: it showed an inverse outcome value signal, i.e. BOLD activity increased with relative value of the alternative (unchosen minus chosen option, reward minus effort magnitude;yellow, whole-brain cluster-corrected, $p=5*10^{-5}$, GLM1). (**Bii**) Individual differences in this neural signal were predictive of individual differences in how well participants could use the learnt information (red, whole-brain cluster-corrected, $p=2*10^{-5}$, GLM2). (**Biii**) Individual differences in this neural signal also correlated with relative glutamate and GABA levels (blue, cluster-corrected in spectroscopy ROI, p=0.039, GLM3). See *Figure 3—figure supplement 2* for overlaps of these activations with the spectroscopy voxel in sagittal cross-sections. Data for A in Figure3_SourceData1; data for B in Figure3_SourceData2.zip

The following source data and figure supplements are available for figure 3:

**Source data 1.** This table contains the spectroscopy, brain volume and behavioral parameters used for correlations in *Figure 3A*.

**Source data 2.** This folder contains the MRI contrast maps, both thresholded (i.e. corrected for multiple comparison using cluster correction) and non-thresholded.

**Source data 3.** This folder relates to *Figure 3—figure supplement 3*.

**Source data 4.** This table relates to *Figure 3—figure supplement 3*.

**Figure supplement 1.** Correlations between spectroscopy measurements and other behavioral parameters.

**Figure supplement 2.** Overlap between neural signals and spectroscopy voxel placements.

**Figure supplement 3.** Different brain volume normalizations for spectroscopy.

**Figure supplement 4.** Correlations between regressors in GLM1.

**Table 1.** (A) Several areas carried a signal for learnt information (relative reward outcome minus relative effort outcome) as an inverse outcome value signal, in other words a signal related to the value of the alternative choice compared to the value of the action actually taken. (B) Other areas signaled outcomes to be learnt as the value of the action actually taken relative to the value of the alternative action. (C) Areas in which individual differences in the strength of the neural signal for the learnt information correlated with the behavioral use of the learnt information. All results are cluster-corrected at whole-brain level (z > 2.3, p<0.05, with actual p-value and number of voxels in the cluster indicated in the table). Region labels were obtained using atlases in FSL: [1](**Neubert et al., 2015**), [2](**Mars et al., 2011**), [3](**Sallet et al., 2013**), [4](**Mori et al., 2005**).

A) Learnt information (inverse value signal)

| | x | y | z | z-score | Voxels | p-value |
|---|---|---|---|---|---|---|
| dACC (area 8m, anterior rostro-cingulate zone[1]) | 6 | 32 | 36 | 3.62 | 821 | $5*10^{-5}$ |
| Parietal (IPL-D, IPL-C[2]), left | −52 | −58 | 42 | 3.68 | 855 | $3*10^{-5}$ |
| Parietal (IPL-C, IPL-D[2]), right | 52 | −46 | 42 | 3.98 | 753 | $1*10^{-4}$ |
| Dorsolateral prefrontal cortex (area 9/46 V[3]), right | 40 | 22 | 38 | 3.61 | 840 | $4*10^{-5}$ |
| Cerebellum | −10 | −80 | −26 | 4.19 | 454 | $6*10^{-3}$ |
| Lateral frontal pole[1], right | 32 | 54 | 6 | 3.21 | 360 | 0.02 |
| B) Learnt information (outcome value signal) | | | | | | |
| Temporal cortex, extending to parietal opercular cortex, left | −36 | −32 | 16 | 3.49 | 1371 | $1*10^{-7}$ |
| Temporal cortex, extending to parietal opercular cortex, right | 50 | −28 | 26 | 3.23 | 458 | $5*10^{-3}$ |
| C) Brain behavior interaction for learnt information | | | | | | |
| Midcingulate cortex (posterior rostro-cingulate zone1) | 2 | −4 | 54 | 3.12 | 719 | $2*10^{-4}$ |
| Pre-SMA extending into dACC and area 8 m[1] | −14 | 24 | 58 | 3.44 | 771 | $2*10^{-5}$ |
| Occipital lobe | −12 | −84 | 4 | 3.22 | 488 | 0.001 |
| White matter (corticospinal tract[4]) | −18 | −14 | 32 | 3.4 | 431 | 0.003 |
| Precentral gyrus, right | 40 | −14 | 52 | 3.28 | 311 | 0.02 |

information learnt from the outcome of one decision to guide subsequent decisions. In contrast, in other situations, it may be beneficial to behave more randomly (for example when exploring new environments). Here, increased GABA concentrations might enable better performance by ensuring that one does not rely too much on previous information. In fact, inhibition in ACC of rats has been shown to disable reward history-guided behaviors, making them more random, which depending on the task led to better or worse performance, similarly inactivation of dACC in macaques completely prevented them from using reward history (**Kennerley et al., 2006**; **Amiez et al., 2006**; **Karlsson et al., 2012**; **Tervo et al., 2014**). It is possible that transient inhibition (through increased GABA) might allow for learning a new model of the task, whereas glutamate might mediate the exploitation of such a model.

DACC has also been implicated in error monitoring. In this context, global changes in another neurotransmitter, acetylcholine, have been shown to affect dACC-mediated post-error adjustments (**Danielmeier et al., 2015**). This suggests that there are additional neurochemical factors, potentially mediating dACC's impact on neural activity in other brain areas.

Our results contrast with findings in vmPFC where increased GABA levels are linked to improved decision accuracy and slower ramping of neural signals (**Jocham et al., 2012**). Here we found that both decreased levels of GABA and increased levels of glutamate were related to the degree to which a learned task model, as opposed to information displayed on each trial, influenced behavior. This suggests a fundamental difference in function, that dACC represents and regulates the use of a model of the world based on past experiences, rather than that it mediates the integration and selection of all arbitrary types of information during decisions (**Hunt et al., 2015**). It is particularly in complex environments that monitoring and fine-tuning of how much to use learnt information – as opposed to immediately perceived information - may be crucial.

These findings are of potential clinical relevance as, dACC has been linked to psychiatric disorders generally (**Goodkind et al., 2015**) and to mood disorders more specifically (**Yüksel and Öngür,**

2010). In the future, it would be important to test whether glutamate and GABA measurements, and their effects on value-guided learning, are changed in mood disorders.

## Materials and methods

### Participants

30 healthy volunteers took part in the study after giving informed consent. One participant was excluded because he/she fell asleep, one participant was excluded because of corrupted spectroscopy data and one participant was excluded because of noise in the spectroscopy measurements (i.e. Cramer-Rao lower bound values for GABA were 38%). Of the remaining 27 participants, 13 were assigned to a selective serotonergic re-uptake inhibitor for two weeks, while 14 were assigned to placebo as part of previously reported studies (*Scholl et al., 2015*). The drug manipulation had no effect on neurotransmitter levels (p>0.84). Nevertheless, we included it in all analyses as a confound regressor.

### Task description

This task description is adapted from a previously published study based on the same task (*Scholl et al., 2015*). We designed a task that allowed measuring how participants learnt about reward and effort and how well they could use this information to guide decisions. In the task, participants made repeated choices between two options with the aim of maximizing their monetary pay-off and minimizing the effort they needed to exert in an interleaved 'effort phase'. On each trial, there were three phases: first participants chose between two options ('choice phase'), then they were shown the outcome of their choice ('outcome phase'), then they had to exert the effort associated with the option they had chosen ('effort phase').

In the decision phase, participants chose between two options using two buttons on a trackball mouse. Each option had three independent attributes: a reward magnitude (reward points, later translated into monetary pay-off), an effort magnitude (amount of effort required in the effort phase), and a probability of receiving a real reward (rather than a hypothetical reward, see below). The probability of each option was shown on the screen at the time of choice. In contrast, the reward and effort magnitudes of the options were not explicitly instructed and instead participants had to learn and track these slowly changing features of the two choices across trials. These magnitudes were drawn from normal distributions of which the means fluctuated pseudorandomly, slowly and independently over the course of the experiment between three levels (low, mid, high). Participants were instructed to learn and keep track of the changing mean value of each magnitude across the experiment. Only one of the reward or effort magnitude means was drifting at any one time and each of the four magnitudes was at each mean level equally often.

After the participants had selected an option, it was highlighted until the ensuing outcome phase. In the outcome phase, participants were first shown the reward and effort magnitudes of the option they had chosen, as well as whether they received a reward or not (in other words whether the outcome was a real secondary reinforcer indicating a specific monetary payment or instead hypothetical). If they received a reward, the current trial's chosen reward magnitude was added to their total reward accumulated so far (which was translated into a monetary reward in the end of the experiment). They were then shown the reward and effort magnitudes for the option they had not chosen. During the outcome phase, participants could thus use the information displayed to update their estimates of the reward and effort magnitudes associated with the choices.

Finally, on every trial, participants had to perform the effort phase of the trial. Participants had to exert a sustained effort by selecting circles that appeared on the screen using the trackball mouse. The circles were added to random positions on the screen in threes every three seconds (up to a total equal to the chosen effort magnitude). To make the task more effortful a random jitter (five pixels, the total screen size was $1280 \times 800$ pixels) was added to the mouse movement and circles only had a 70% probability of disappearing when clicked on. Furthermore, we pre-screened participants and only invited participants for the fMRI session if they had perceived the effort as aversive and were willing to trade-off money to reduce the effort that they needed to exert.

Participants had 25s to complete the clicking phase and otherwise lost money equivalent to the potential reward magnitude of the chosen option (participants failed to complete the effort phase on less than 1% of trials).

On most trials (100 out of 120) participants had to chose between the two options with changing reward and effort magnitudes. The reward magnitudes were set between 0 and 20 pence and the effort magnitudes were set between 0 and 15 circles that needed to be clicked. On the remaining trials ('Special-option-trials', SOTs), participants had to choose between one of the changing options and one of two fixed options whose values participants learned in a training session outside the scanner. The value of both fixed options was 7.5 pence, but one had a fixed effort magnitude of 4 circles and the other had one of 12 circles. The SOTs were included to ensure participants learned the values of each choice, rather than just their preference for one option over the other (a relative preference for one option over the other would not enable participants to choose effectively on the SOTs).

Interspersed with the 120 learning trials, there were 20 trials on which participants just had to indicate which option had a higher mean effort magnitude. These trials were included to ensure participants paid attention to the effort dimension. They were not given feedback about their choice. These trials were not included in the data analysis.

Participants performed 120 trials of the learning task inside the scanner and an additional 120 trials afterwards on the next day outside the scanner to increase the number of trials for the behavioral data analysis. Each participant performed the same two schedules in randomized order. Participants were informed about the features of the task in two training sessions before the scan, including the fixed number of trials they would perform. This ensured that they did not perceive low effort options as having a potentially higher monetary value because taking them might allow participants to move on to the next trial more quickly and to perform more trials with more chances to win money. Further details of the training were as follows: In the first training session (45 min), participants performed a version of the task without a learning component, i.e. not only the probability, but also reward and effort magnitudes were explicitly shown. This training ensured that participants were familiar with the features of the task, for example, that they understood what the probability information meant. We also used this session to exclude participants before the fMRI session that did not find the effort sufficiently aversive to produce robust effects on behavior. In a second training session (1 hr), we instructed participants about the learning task that they later performed in the fMRI scanner. At the end of the training, participants were queried about how they made decisions (specifically, they were asked 'What are you thinking about when you're making your decision'). All participants reported trying to learn the reward and effort magnitudes and using the explicitly cued probabilities to make decisions. This suggested that participants were well aware of how to do the task before the beginning of the scan.

## Experiment timings

The options were displayed for 1.4 to 4.5 s before participants could make a choice. After the choice was made, the chosen option was highlighted for 2.9 to 8.0 s. Next, the outcome was first displayed for the chosen option (1.9–2.1 s), then for the unchosen option (1.9–6.9 s). Participants then performed the effort exertion task (0–25 s). Finally, the trial ended with an ITI (2.3–7.5 s).

## Data sharing

The data are publicly available from the Oxford University Research Archive (https://doi.org/10. 5287/bodleian:PP805bgDz). Analysis scripts are available on request from the corresponding author. Source data files are provided with the article for all figures presented in the manuscript.

## Task validation

We performed a logistic regression to validate that participants performed the task well, i.e. that they took all relevant task features into account when making their decisions. In the regression, we predicted whether participants chose again the same option as on the previous trial ('stay') or instead selected the alternative option ('switch'). As predictors we included the displayed reward probabilities (from current trial, t) and the reward ('RM') and effort magnitudes ('EM') from the past four trials (t-1, t-2, t-3, t-4). These regressors were coded in the frame of reference of the 'stay'

choice relative to the 'switch' choice [e.g. reward magnitude on the last trial (t-1) for the option that would be a 'stay choice' minus the reward magnitude (at t-1) for the alternative option]. All regressors were z-score normalized.

$$Y = \beta_0 + \beta_1 Reward\ Probability_t + \beta_2 RM_{t-1} + \beta_3 RM_{t-2} + \beta_4 RM_{t-3} + \beta_5 RM_{t-4}$$
$$+ \beta_6 EM_{t-1} + \beta_7 EM_{t-2} + \beta_8 EM_{t-3} + \beta_9 EM_{t-4}$$

We used ANOVAs to test whether participants could use the learnt information (main effect across the four reward magnitude (RM) or the four effort magnitude (EM) regression weights). We controlled for group assignment as a between participant confound.

The same result can be illustrated by binning participants' choices according to the predicted reward and effort magnitudes on each trial, as derived from a previously established Bayesian optimal observer model (*Scholl et al., 2015*), see also *Figure 2—figure supplement 2A* and the Materials and methods below for a validation of this model.

## Behavioral modeling

We adapted a previously described computational learning model (*Scholl et al., 2015*) to measure how much participants used the information they learnt to guide their choices (γ). This Rescorla-Wagner learning model was fit to participants' choices in the task. In short, the model consisted of three components: firstly the model had predictions of the mean reward/effort magnitudes underlying both outcomes. These were updated on every trial:

$$Prediction_t = Prediction_{t-1} + \alpha * PE_{t-1}$$

with

$$PE_{t-1} = Outcome_{t-1} - Prediction_{t-1}$$

where α was the learning rate.

Secondly, the model combined these reward/effort magnitude predictions together with the reward probabilities (shown to participants on each trial) to calculate how valuable each option was (i.e. their utility).

$$Utility_{OptionA} = (1 - \gamma) * Probability_{Reward} + \gamma * (\lambda * MagPrediction_{Reward}$$
$$- (1 - \lambda) * MagPrediction_{Effort})$$

λ describes to what extent participants relied more on reward versus effort magnitudes. In the present context, the parameter of interest that describes how much participants used the learnt information was γ.

Thirdly, the model then compared the utility of the two options to predict choices, using a standard soft-max decision rule:

$$P(Option_A) = \frac{e^{\beta * Utility_A}}{e^{\beta * Utility_A} + e^{\beta * Utility_B}}$$

where $\beta$ (inverse temperature) reflected participants' tendency to pick the option with the higher utility.

We also considered alternative models (M2-M6, see *Figure 2—figure supplement 2A*). Firstly, these models differed in their number of learning rates: they either shared the same learning rate for reward and effort, or they had separate learning rates. Secondly, instead of computing utility as a linear combination of reward magnitudes and probability, utility could be computed based on a multiplicative integration of probability and reward:

$$Utility_{OptionA} = (1 - \lambda) * Probability_{Reward} * MagPrediction_{Reward} - \lambda$$
$$* MagPrediction_{Effort}$$

where λ was the relative effort (to reward) sensitivity.

Finally, to ensure that the previously described (*Scholl et al., 2015*) Bayesian optimal observer model that we used to illustrate the participants' behavior in *Figure 2A* and to derive regressors for the fMRI analysis provided a good fit to the data, we also used a model with no fitted learning rate

that instead used the predictions for reward and effort derived from the Bayesian model. We also note here that fMRI regressors derived from the Bayesian optimal observer model correlated very strongly (r > 0.99) with those obtained from the fitted reinforcement learning model and therefore using either type of model to obtain regressors does not affect our results.

All models were fit using Bayesian parameter estimation (*Lee and Wagenmakers, 2014*) as implemented in Stan (*Carpenter et al., 2016*). We used a hierarchical modeling approach, i.e. parameter estimates for individual participants were constrained by a group-level distribution of those parameters. We obtained three chains of 1000 samples after an initial warm-up of 1000 samples; convergence of chains was checked (*Gelman and Rubin, 1992*). Based on an initial fitting of individual participants, parameter ranges and transformations were selected so that parameters were reasonable, and normally distributed at the group level. Specifically, the learning rates, weight of learnt information and sensitivity to the effort were sampled from a group normal distribution on a scale from -∞ to +∞ and then transformed to a scale from 0 to 1; the inverse temperature was sampled from a group normal distribution on a scale from 0 to 1. For the group level distributions, mean values were given a flat prior in the allowed range and standard deviations were given a prior of mean zero and standard deviation 10 and constrained to be positive.

We assessed model-fit in two ways. Firstly, we performed a cross validation using a half-split of the data: we fitted all participants' data for either the session inside or outside the scanner and then used the estimated parameters to assess predictive accuracy (summed log likelihoods) for the data from the other session (*Vehtari et al., 2016*). Secondly, as an alternative method for model comparison, we also computed summed (across participants) BIC values for a non-hierarchical version of the models, fitted using Matlab's fminsearch. We also used the parameter estimates derived from the separate sessions to examine test-retest reliability of the parameter estimates (*Figure 2—figure supplement 2C*).

In supplementary analyses, we validated the model (M1) further (*Figure 2—figure supplement 1*). To check that our model was indeed able to capture participants' behavior, we simulated data from 10 sets of 27 participants with parameter group mean and standard deviations as derived from the real data. We analyzed this simulated data using the same regression and model-fitting approaches as described above. To illustrate our behavioral effect of interest, differences in the use of learnt information (γ), we simulated another two groups of 270 participants whose mean γ was at the extreme ends of the confidence intervals for those found in real participants.

## Spectroscopy

Spectroscopy and fMRI data were acquired using a Siemens Verio 3 Tesla MRI scanner (32-channel coil). Spectroscopy data were obtained from dACC. Previous studies have shown that spectroscopy measurements of neurotransmitter levels are region specific (*Emir et al., 2012*; *van der Veen and Shen, 2013*). First, a high-resolution T1-weighted scan was acquired using an MPRAGE sequence. Based on this scan, the spectroscopy voxel (2 x 2 × 2 cm) was centered on dACC by reference to the location of the corpus callosum, the cingulate and adjacent sulci. The center of gravity of the region of maximum voxel overlap across participants lay at x = 0, y = 28, z = 28 in the Montreal Neurological Institute (MNI) space. The relatively large size of the spectroscopy voxel meant that it extended to include tissue in the paracingulate sulcus in those participants in which it was present. MRS data (128 samples) were acquired using the SPECIAL sequence (*Mekle et al., 2009*; *Mlynárik et al., 2006*) as described previously (*Stagg et al., 2011*). The data were preprocessed using the FID-Appliance (github.com/CIC-methods/FID-A [*Simpson et al., 2015*]) to correct for frequency and phase-drift. The data were then analyzed using LCModel (*Provencher, 2001*). Voxels for which Cramer-Rao lower bound values exceeded 20% were excluded. GABA and glutamate values were divided by total creatine. To correct for partial volumes within the spectroscopy voxels, all analyses included as confound regressors the relative volumes of grey and white matter (i.e. grey or white matter divided by total tissue = grey + white + cerebrospinal fluid) and total tissue in the spectroscopy voxel. These values were obtained using FAST (FMRIB's automated segmentation tool, [*Zhang et al., 2001*]). The results were independent of the precise manner of controlling for partial volume (see *Figure 3—figure supplement 3*).

## Relating behavior to spectroscopy

As in previous reports (*Jocham et al., 2012*), we found that glutamate and GABA were correlated (r = 0.47, p=0.013). Therefore, to be able to measure the separate impact of glutamate and GABA on the use of the learnt information, we performed nonparametric (Spearman) partial correlations, between the use of learnt information (γ) and either glutamate or GABA, controlling for the other neurotransmitter. In each analysis, we additionally controlled for group assignment, inverse temperature (β, from the behavioral model), relative gray and white matter and total tissue. To reduce the number of multiple comparisons in our initial analysis, when testing each model parameter for its relationship to glutamate and GABA, we combined glutamate and GABA to one value (glutamate minus GABA). This also reflected our initial hypothesis that it might not be each neurotransmitter in isolation that influences neural activity and behavior but rather the relationship between glutamate and GABA that is critical.

## FMRI

### Data acquisition

For the fMRI, we used a Deichmann echo-planar imaging (EPI) sequence (*Deichmann et al., 2003*) [time to repeat (TR): 3s; 3 x 3 × 3 mm voxel size; echo time (TE): 30 ms; flip angle: 87°; slice angle of 15° with local z-shimming] to minimize signal distortions in orbitofrontal brain areas. This entailed orienting the field-of-view at approximately 30° with respect to the AC-PC line. We acquired between 1100 and 1300 volumes (depending on the time needed to complete the task) of 45 slices per participant.

### Preprocessing

FMRI data were analyzed using FMRIB's Software Library (FSL [*Smith et al., 2004*]; see also [*Scholl et al., 2015*]), run on a local computer using HTCondor (*Thain et al., 2005*) and code from NeuroDebian (*Halchenko and Hanke, 2012*). We used the standard settings in FSL (*Smith et al., 2004*) for image pre-processing and analysis. Motion was corrected using the FSL tool MCFLIRT (*Jenkinson et al., 2002*). This also provided six motion regressors that we included in the FMRI analyses. Functional images were first spatially smoothed (Gaussian kernel with 5 mm full-width half-maximum) and temporally high-pass filtered (3 dB cut-off of 100 s). Afterward, the functional data were manually denoised using probabilistic independent component analysis (*Beckmann and Smith, 2004*), visually identifying and regressing out obvious noise components (*Kelly et al., 2010*); we considered only the first 40 components of each participant which had the greatest impact to interfere with task data (total up to 550). We used the Brain Extraction Tool (BET) from FSL (*Smith, 2002*) on the high-resolution structural MRI images to separate brain matter from non-brain matter. The resulting images guided registration of functional images in the Montreal Neurological Institute (MNI)-space using non-linear registrations as implemented in FNIRT (*Jenkinson et al., 2012*). The data were pre-whitened before analysis to account for temporal autocorrelations (*Woolrich et al., 2001*).

### Data analysis

In the first analysis (GLM1), we looked for brain areas that showed activity varying with the reward and effort information to be learnt. A full list of regressors and correlations between them is shown in *Figure 3—figure supplement 4* (all r<0.33). We used three boxcar regressors, indicating the onset and duration of the decision phase (from the beginning of the trial until participants made a choice), the onset and duration of the outcome phase (from the appearance of the chosen outcome until the chosen and the unchosen outcomes disappeared from the screen) and lastly the effort exertion phase (from the appearance of the first effort target until participants had removed the last target). In the outcome phase, we included the following parametric regressors: whether a reward was delivered for the chosen option, the reward probability for the chosen option and the reward and effort magnitude outcomes for the chosen and the unchosen option. In each case, separate regressors for the chosen and the unchosen option were used.

The main contrast of interest (*Figure 3B*) was the total information to be learnt, i.e. the contrast of the relative (chosen minus unchosen) reward minus effort magnitude:

$$Learnt\ Information_{Contrast} =$$
$$(RewardMagnitude_{Chosen} - RewardMagnitude_{Unchosen}) -$$
$$(EffortMagnitude_{Chosen} - Effort_{Unchosen})$$

We used FSL's FLAME 1 + 2 (*Woolrich et al., 2004*) to perform higher-level analyses; outlier de-weighting was used. We included group assignment as a confound regressor. Results were cluster-corrected (p<0.05, voxel inclusion threshold: z = 2.3).

Next, we tested whether individual differences in how much participants could use the learnt information related to differences in neural signals (GLM2). For this, we included at the group level the behavioral measure γ as a covariate. We again included group assignment, as well as inverse temperature (β) as confound regressors. The results were cluster-corrected (p<0.05, voxel inclusion threshold: z = 2.3).

### Relating fMRI to spectroscopy measures

We tested how measures of GABA and glutamate influenced the neural signal of the information to be learnt (GLM3). For this, we included at the group level GABA and glutamate measurements as covariates. As confound regressors we included, as in the behavioral analysis, group assignment, inverse temperature (β), as well as gray matter (voxel-wise, obtained using FSL's feat_gm_prepare), relative white matter and total tissue in the spectroscopy voxel. We combined regressors for the effect of glutamate and GABA to a single contrast for statistical testing (i.e. glutamate minus GABA levels). We used the group average spectroscopy voxel as a mask; results were again cluster-corrected (p<0.05, voxel inclusion threshold: z = 2.3).

## Acknowledgement

The authors thank Gerhard Jocham for helpful advice on methods and data analysis.

## Additional information

### Funding

| Funder | Grant reference number | Author |
|---|---|---|
| Medical Research Council | MR/N014448/1 | Jacqueline Scholl |
| Wellcome Trust | 092759/Z/10/Z | Jacqueline Scholl |
| Wellcome Trust | 089280/Z/09/Z | Nils Kolling |
| Wellcome Trust | WT100973AIA | Matthew FS Rushworth |
| Christ Church College | Stipendary Junior Research Fellowship | Nils Kolling |

The funders had no role in study design, data collection and interpretation, or the decision to submit the work for publication.

### Author contributions

JS, NK, Conception and design, Acquisition of data, Analysis and interpretation of data, Drafting or revising the article; NN, CJS, Acquisition of data, Drafting or revising the article; CJH, MFSR, Conception and design, Analysis and interpretation of data, Drafting or revising the article

### Author ORCIDs

Jacqueline Scholl, http://orcid.org/0000-0001-9969-1355

### Ethics

Human subjects: Participants gave informed consent to take part in the study, which was approved by the NRES Committee South Central - Portsmouth (12/SC/0276)

## Additional files

### Major datasets

The following dataset was generated:

| Author(s) | Year | Dataset title | Dataset URL | Database, license, and accessibility information |
|---|---|---|---|---|
| Scholl J, Kolling N, Rushworth M, Harmer C | 2016 | Neurotransmitters in learning and decision-making - Data | https://ora.ox.ac.uk/objects/uuid:bde6944b-3021-4942-a2cb-9f76fef12788 | Publicly available at the Oxford University Research Archive (https://ora.ox.ac.uk) |

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
