## [Decision Letter]

Thank you for submitting your article "Excitation and inhibition in dorsal anterior cingulate predict brain activity and use of past experiences" for consideration by *eLife*. Your article has been favorably evaluated by David Van Essen as the Senior Editor and three reviewers, including Emmanuel Procyk (Reviewer #3) and Joshua Gold, who is a member of our Board of Reviewing Editors.

The reviewers have discussed the reviews with one another and the Reviewing Editor has drafted this decision to help you prepare a revised submission.

Summary:

This study tested whether levels of Glu and GABA, as measured by MR spectroscopy in dorsal anterior cingulate cortex (dACC), correlate with brain function and behavioral markers attributed to dACC. They first made these measures in the rostral part of the mid-cingulate cortex, taking into account sulcal morphology. The defined ROI in the region served for spectroscopic measurements and for comparisons with BOLD measurements obtained while subjects performed a multidimensional learning task. The task tested the degree to which the subjects used outcome information to adapt decisions to choose between alternatives. A model-based approach, validated by models comparisons, allowed the authors to fit subjects' behavior and extract a parameter of interest describing the tendency of subjects to use learnt information.

The authors present three primary findings: 1) dACC levels of Glu were positively correlated and of GABA were negatively correlated with the degree of experience-based learning on a subject-by-subject basis; 2) a partly overlapping brain area represented the information to be learned in a manner that was also correlated with the behavioral effect; and 3) there was also a relationship between the strength of this representation and neurotransmitter levels. In addition, this study replicated previous work with the multidimensional learning task, thereby strengthening the notion that dACC mediates updating of cognitive models used to drive adaptive reward-guided behavior.

The reviewers were in agreement that this study is interesting and relevant, in that it addresses mechanisms of adaptive decisions, which is a critical aspect of higher brain function. In addition, the methodology is sound, and the paper is clearly written.

Essential revisions:

1) The task has some nice features, including the use of explicit and learned information that can be used flexibly to guide behavior. However, it also seems awfully complicated to allow for a single parameter to effectively describe the overall influence of learned information on behavior. Some issues that this raises: 1) is the learning parameter even consistent for a single subject over time? What is the test/retest reliability? 2) Were there systematic relationships between how well the model fit behavior for individual subjects and the MRS/fMRI activity?

2) The main results obviously depend on statistical outcomes that hopefully are not overly sensitive to multiple comparisons and other factors. Was that tested? The glutamate result presented in Figure 3 seems particularly weak; does it survive a non-parametric test? Did the neurotransmitter measures relate to any of the other behavioral parameters?

3) It would be useful to discuss in more detail the interpretation of the findings in terms of the role of dACC in adaptive behavior. For example, why it is expected that dACC bold inversely correlates with the information to be learnt, and how does this finding logically lead to the negative correlation with GABA? The answer to this might explain why the testing of alternative areas concerned only the deactivated network (Table 1). On a related note, are there more specific points to be made regarding the meaning of the sign of the "information to be learned" relative difference? Likewise, the results suggest that more GABA in dACC leads to less use of information to guide behavior. How does this relate to more random behavior found in rats? More generally, a more mechanistic interpretation of the role of GABA/GLU in reducing integration of reward-history (or other behaviorally relevant computations) would be useful.

---

## [Author Response]

*Essential revisions:*

*1) The task has some nice features, including the use of explicit and learned information that can be used flexibly to guide behavior. However, it also seems awfully complicated to allow for a single parameter to effectively describe the overall influence of learned information on behavior.*

We agree with the reviewer that our experiment is more complex than most learning tasks. This is because we wished to investigate specific aspects of behavior. However, we believe that we can capture the key component of using learned information as opposed to instructed information with one specific parameter that we can separate from the other major determinants of behavior. We have been able to measure this component independently of learning speeds, for rewards or for costs; inverse temperature, and sensitivity to cost vs. rewards.

The reviewer is of course also correct in pointing out that in such a complex task it is important to test whether the model fits reliably. We have now expanded the manuscript with two further figures (Figure 2—figure supplement 1) that assess appropriateness of the model, model fit and reliability, and which illustrate the modeling approach better.

In a new supplementary figure (Figure 2—figure supplement 1), we simulated agents that make decisions in exactly the same task as our participants, with behavior being guided by the same computational model (M1) that we use throughout the paper. Importantly, these simulated agents value the different choices according to a weighted sum of learnt information and cued information that is traded-off by a single parameter, the use of learnt information, γ. We find that these simulated agents behave just like our human participants, as revealed by a regression analysis (Figure 2—figure supplement 1) analogous to the one performed on human participants’ data (Figure 2). We also find that when we then use our computational model to analyze the simulated participants’ data, we can recover the model parameters well (Figure 2—figure supplement 1). This further suggests that a model that describes the overall influence of learnt information with a single parameter is appropriate for our task.

We have now also added an illustration of how a change in the use of learnt information will affect behavior (Figure 2—figure supplement 1): We simulated agents that differ in the value of their parameter determining the use of learnt information. We find, when binning the simulated choices by either the learnt value or the explicitly cued value, that when γ is higher, behavior is, as expected, more driven by the learnt information. In contrast, when γ is lower, behavior is more drive by differences between the options in terms of their cued value (i.e. explicitly shown probabilities).

We have expanded the Methods (section ‘Behavioral Modeling’) in the following way to reflect these considerations:

“In supplementary analyses, we validated the model (M1) further (Figure 2—figure supplement 1). […] To illustrate our behavioral effect of interest, differences in the use of learnt information (γ), we simulated another two groups of 270 participants whose mean γ was at the extreme ends of the confidence intervals for those found in real participants.”

We have also included a new supplementary figure showing the new results of model validation (Figure 2—figure supplement 1).

*Some issues that this raises: 1) is the learning parameter even consistent for a single subject over time? What is the test/retest reliability?*

We have carried out a comparison of the learning parameter (use of learnt information γ), and other parameters used to describe our participants’ behavior, across time periods as requested by the reviewer. In short, we find that parameters are indeed very consistent over time.

Our behavioral data was collected in two separate sessions, one inside the MRI scanner, while we collected fMRI data (120 trials) and one outside the scanner, on the next day (120 trials). We can therefore examine the test/retest reliability of parameter estimates based on either the 120 trials inside or the 120 trials outside the scanner (collected on the next day). To do this, we have changed our modeling approach to employ a hierarchical Bayesian model fitting method as implemented in Stan (Carpenter et al., 2016), as this has been suggested to provide more robust fits when less data per participant is available (a detailed description of this new fitting approach has been included in the manuscript and is also shown below). First, we note that the parameters, based on data from both sessions, obtained using either the hierarchical or the non-hierarchical approach that we used before are very highly correlated (see Figure 4).

Author response image 1.Correlations between model parameters derived using either a hierarchical (y-axis) or non-hierarchical (x-axis) model fitting approach.**DOI:**
http://dx.doi.org/10.7554/eLife.20365.020

Second, using the hierarchical fitting approach on data from each session separately, we find (Figure 2—figure supplement 2, a strong correlation between the parameter estimates for the two sessions for our main parameter of interest, γ, how much participants used what they had learnt, and also all other parameters of the model. We therefore concluded that our model parameters are reliable and robust within individual subjects across time (at least from one day to the next).

We also note that as this form of test/retest reliability was only done on half the data, it seems likely that parameters fitted on the whole data set (which is the approach that we use in the main paper) would be estimated even more precisely and therefore show even higher test/re-test reliability.

We have expanded the Methods (section ‘Behavioral modeling’) to describe this new hierarchical model fitting approach:

“All models were fit using Bayesian parameter estimation (Lee and Wagenmakers, 2014) as implemented in Stan (Carpenter et al., 2016). We used a hierarchical modeling approach, i.e. parameter estimates for individual participants were constrained by a group-level distribution of those parameters. We obtained three chains of 1000 samples after an initial warm-up of […] We also used the parameter estimates derived from the separate sessions to examined test-retest reliability of the parameter estimates (Figure 2—figure supplement 2).”

We have included a new supplementary figure to show the test/re-test reliability (Figure 2—figure supplement 2 panel C).

*2) Were there systematic relationships between how well the model fit behavior for individual subjects and the MRS/fMRI activity?*

In short, there were no relationships between model fit and our measures of interest. Additionally, controlling for a proxy of individual differences in model fit in our analyses does not affect any of our results.

Using model-fitting based on individual participants (i.e. without a hierarchical model), we obtain one measure of model fit for each participant. This correlates strongly with the estimated inverse temperature, i.e. an index of behavioral stochasticity according to our model, see Figure 5. The overall goodness of model fit does not, however, correlate with our parameter of interest, γ, the use of learnt information, showing that an overall non-specific signal strength that correlates with participants’ willingness or ability to do the task cannot be invoked to explain our results.

Author response image 2.The fit of the model for each participant (measured as negative log likelihood, y-axis) strongly correlates with the estimate of inverse temperature (β) from the model (ρ=-0.78, p=3.5x10^-6^).However it does not correlate with other model parameters (all p>0.12).**DOI:**
http://dx.doi.org/10.7554/eLife.20365.021

Using the new hierarchical fitting method, we no longer obtain a measure of model fit for each person (instead there is only a measure of model fit for all participants together). As a proxy, because of the relationship between inverse temperature and model fit for individual participants (Figure 5), we instead used the inverse temperature. We find that inverse temperature does not correlate with spectroscopy measures (see Figure 3—figure supplement 1). We therefore conclude that it is not the case that individual differences in how much people use what they have learnt are an artifact of overall behavioral model fit. We have now included the inverse temperature as a control parameter in analyses throughout the manuscript. We have updated all figures and tables in the manuscript accordingly.

*2) The main results obviously depend on statistical outcomes that hopefully are not overly sensitive to multiple comparisons and other factors. Was that tested? The glutamate result presented in Figure 3 seems particularly weak; does it survive a non-parametric test? Did the neurotransmitter measures relate to any of the other behavioral parameters?*

In short, we find that also after correction for multiple comparisons and using non-parametric tests, our results hold. Additionally, neurotransmitter measurements do not relate to any other behavioral parameters, but instead are very specifically related to how participants use the learnt information.

Correction for multiple comparisons is not always straightforward, as it is critically dependent on the number of plausible, equivalent i.e. interchangeable tests that would confirm or deny one’s hypothesis. In the analysis of the effects of glutamate and GABA on behavior, our main hypothesis was, based on previous work, that this would relate to the use of learnt information (Karlsson et al., 2014). For this reason we did not examine correlations between neurotransmitter levels and all other behavioral parameters.

However, to show the specificity of our results, we have now also examined these additional correlations and find that none of the other behavioral measures correlate with neurotransmitter measurements. Moreover, the correlation between γ and neurotransmitters remains significant even after correction for multiple comparison (see Figure 3—figure supplement 1; for the use of learnt information: ρ=0.53, p=0.010; for all other parameters, p>0.7). Specifically, to reduce the number of comparisons, we have combined the glutamate and GABA measurement (as z-score normalized glutamate minus GABA levels, as for the fMRI) and performed partial correlations between this value and the four behavioral parameters (use of learnt information, inverse temperature, relative reliance on learnt reward or effort information and learning rate). These four comparisons thus have a Bonferroni-corrected p-value of 0.0125 (0.05/4). Furthermore, all these results are now reported as nonparametric partial correlations (Spearman’s rho, ρ; this has also been implemented in Figure 3 in the main manuscript). The fMRI analyses are controlled for multiple-comparison at the cluster level. In the fMRI analysis, we used FSL’s outlier de-weighting (in FLAME 1+2), which means that the possible impact of outlier data points is reduced. Therefore, our results are robust to correction for multiple comparison and outliers.

We also note (see Figure 3—figure supplement 3) that our results were not sensitive to the precise method of correction of spectroscopy values for brain volumes: we present in our paper results from partial correlation analyses that treat partial grey and white matter brain volume in the spectroscopy voxel as confound regressors. If instead we use other correction methods, we find very similar results. This further attests to the robustness of our results.

*3) It would be useful to discuss in more detail the interpretation of the findings in terms of the role of dACC in adaptive behavior. For example, why it is expected that dACC bold inversely correlates with the information to be learnt, and how does this finding logically lead to the negative correlation with GABA? The answer to this might explain why the testing of alternative areas concerned only the deactivated network (Table 1). On a related note, are there more specific points to be made regarding the meaning of the sign of the "information to be learned" relative difference?*

We are happy to expand our Discussion and we have added a new conceptual figure (Figure 1—figure supplement 1), to help clarify these issues.

We agree that referring to the activity pattern that we found as “deactivation” is potentially confusing and we have made changes throughout our manuscript to make our argument clearer: Similar to our BOLD effect in dACC, previous fMRI studies have found activity in dACC reflects what are sometimes called “inverse value” signals. For example, Kurniawan et al. (2013), Prevost et al. (2010) and Skvortsova et al. (2014) have all reported that dACC activity increases with increasing effort levels and decreases with increased levels of reward, which results in a signal that reports the opposite/inverse of the subjective value of the chosen option. Such a signal can, for the sake of brevity, be described as a “deactivation” in proportion to subjective value of the choice taken as opposed to the choice not taken. However, we agree with the reviewers that such a description is potentially confusing and arguably the activity pattern becomes more intuitive if it is described as positively related to the value of the alternative choice rather than the choice taken. Because the values of the two choices are varied independently we know that the activity is definitely also positively related to the value of the unchosen option. A number of fMRI and single neuron recording studies have suggested that such activity in dACC is related to the weighing up of evidence for changing or maintaining behavior (Kolling et al., 2012; Kolling et al., 2014; Kolling et al., 2016a; Kolling et al., 2016b; Meder et al., 2016; O'Reilly et al., 2013; Scholl et al., 2015; Shima and Tanji, 1998; Stoll et al., 2016; Wittmann et al., 2016). Such an interpretation is certainly consistent with the data in our current experiment.

In short, previous literature and the current study suggest that there is a relationship between dACC and the value of switching to an alternative behavior as opposed to the action actually taken. In the revised manuscript we have tried to express this without using the word “deactivation” which carries entirely inappropriate connotations of an overall decrement in activity during task performance. To take this to the domain of neurotransmitters, an enhanced learning signal in dACC would mean a stronger signal indicating the relative value of information to be learnt for an alternative as opposed to the current behavior. If glutamate increases a signal and GABA decreases it then we should see a larger instance of such a signal as glutamate increases and as GABA decreases. This is exactly what we find.

Regarding the choice of control regions. The aim of these control analyses was to assess whether the neurotransmitter levels measured in dACC specifically predicted BOLD signal in dACC or also BOLD signal in other brain areas. We did this as an additional control, despite evidence that glutamate and GABA levels in different brain areas are relatively uncorrelated (Emir et al., 2012; van der Veen and Shen, 2013). In order to perform the most stringent control analysis, we chose control areas, which were identified as having activity levels that changed in the same manner and direction as dACC in the same contrast (information to be learnt). However, we agree with the reviewer that we could have also examined activity in areas that showed the opposite pattern of activity in relation to our contrast of interest (i.e. the areas listed in Table 1). We have now performed this additional control analysis, identifying regions of interest in mid cingulate cortex and temporal cortex, extending into parietal cortex. Again, we find no evidence that dACC spectroscopy measures correlate with activity relating to information to be learnt in these other brain regions suggesting, once again, the specificity of our effect.

We have now updated the manuscript to reflect these considerations. We have included a new conceptual figure (Figure 1—figure supplement 1).

We have also reworded relevant parts in the Results section to avoid the term ‘deactivation’:

“We identified activity in dACC and adjacent cortex (Figure 3Bi, x=6, y=32, z=36, z-score = 3.62, cluster p-value=5x10^-5^) and in other areas (Table 1) as coding the information to be learnt as an inverse outcome value signal (relative reward outcome minus relative effort outcome) or, in other words, a signal related to the relative value of the alternative not chosen on the current trial. Such a signal has previously been noted in dACC and has been related to behavioral adaptation: decisions to maintain or change behavior in diverse contexts (Kolling et al., 2012; Kolling et al., 2016b; Meder et al., 2016; Shima and Tanji, 1998; Stoll et al., 2016).”

We have now also added a note in the Results section to highlight that we have used both types of regions coding information to be learnt, i.e. regions with a value signal in the framework of the chosen or the unchosen option, as control regions:

“This result was specific to dACC; analogous analyses in other ROIs identified in the contrasts for learnt information (Table 1) revealed no significant effects.”

*Likewise, the results suggest that more GABA in dACC leads to less use of information to guide behavior. How does this relate to more random behavior found in rats? More generally, a more mechanistic interpretation of the role of GABA/GLU in reducing integration of reward-history (or other behaviorally relevant computations) would be useful.*

We apologize to the reviewer for not making it clearer how we place our findings mechanistically and in regards with more basic rodent studies. We have now made an additional illustration (Figure 1—figure supplement 1) to hopefully make things clearer and we have included some more explanations in the text.

In short, we interpret our results as suggesting that the glutamate to GABA levels in dACC are a mechanism through which the brain can control to what extends it relies on either information that has been learnt (reward histories) or on new information available at the time of the decision. More excitation could drive increased firing of neurons with reward history based estimates of value, leading behavior to be more influenced by reward histories. GABA on the other hand reduces firing, preventing such information from driving behavior and effectively suppressing the effect of such past experiences on choice.

This relates very directly to findings by Tervo et al. (2014). They found that manipulating activity in rat ACC (through e.g. muscimol inactivation), in other words manipulating inhibitory activity, reduced the extent to which rats based their choices on information they had learnt. In other words it reduced the influence of the rats’ task models on their behavior. In the study of Tervo and colleagues, however, not using learnt information meant animals behaved randomly, whereas in our task they relied more heavily on other features of the task, namely the non-learnt probability information in a trial.

Relatedly, neurophysiological recordings made in ACC by Karlsson et al. (2012) found that ensemble activity patterns related to a model of the world (or prior beliefs) rats had learnt. When animals needed to disregard their learnt model, activity in ACC abruptly changed.

We have now updated the Discussion to reflect these considerations, however because of word limits, we had to keep this quite brief. We also hope that the new Figure 1—figure supplement 1 also illustrates the proposed mechanism better:

“Our findings are consistent with an emerging view of dACC in forming, updating and maintaining a model of the world and of behavioral strategies (Karlsson et al., 2012; Kolling et al., 2014; O'Reilly et al., 2013; Wittmann et al., 2016). […] It is possible that transient inhibition (through increased GABA) might allow for learning a new model of the task, whereas glutamate might mediate the exploitation of such a model.”